# Melatonin Levels in Children with Obesity Are Associated with Metabolic Risk and Inflammatory Parameters

**DOI:** 10.3390/nu13103629

**Published:** 2021-10-16

**Authors:** Marie Gombert, Vanessa Martin-Carbonell, Gonzalo Pin-Arboledas, Joaquín Carrasco-Luna, Álvaro Carrasco-García, Pilar Codoñer-Franch

**Affiliations:** 1Department of Pediatrics, Obstetrics and Gynecology, Universidad de Valencia, 46010 Valencia, Spain; magom@alumni.uv.es (M.G.); vanessamartin7891@gmail.com (V.M.-C.); al.carrasco.garcia@gmail.com (Á.C.-G.); 2Pediatric Sleep Unit, Department of Pediatrics, Quironsalud Hospital, 46010 Valencia, Spain; pinarboledasgonzalo@gmail.com; 3Department of Experimental Sciences, Universitat Católica de Valencia, 46001 Valencia, Spain; joaquin.carrasco@uv.es; 4Service of Pediatrics, University Hospital Doctor Peset, FISABIO, 46017 Valencia, Spain

**Keywords:** melatonin, childhood obesity, metabolism, low-grade inflammation, circadian rhythms

## Abstract

Melatonin, the hormone of circadian rhythm regulation, is involved in the modulation of mitochondrial activity through its antioxidant and anti-inflammatory properties. Alteration of circadian rhythms such as sleep is related to obesity and metabolic pathogenesis in adulthood, but studies during childhood are scarce. The present study investigated the association of melatonin with metabolic and inflammatory markers in children with (*n* = 113) and without obesity (*n* = 117). Melatonin was measured in saliva four and two hours before bedtime, and after one hour of sleep. Cardiometabolic factors, high sensitivity C-reactive protein, immune markers (monocyte chemoattractant protein-1, plasminogen activator inhibitor-1, tumor necrosis α and interferon-γ), leptin and ghrelin were determined. Sleep duration was recorded by a questionnaire. The melatonin level at 1 h after sleep was found to be increased more than twofold in children with obesity (90.16 (57.16–129.16) pg/mL) compared to controls (29.82 (19.05–61.54) pg/mL, *p* < 0.001) and was related to fat mass (rho = 0.294, *p* < 0.001); melatonin levels at 1 h after sleep were inversely correlated with high-density lipoprotein cholesterol. Positive correlation was found with apolipoprotein B, adipokines, high sensitivity C-reactive protein, plasminogen activator inhibitor-1 and tumor necrosis factor-α. Shorter sleep duration and earlier waking times were recorded in children with obesity. In conclusion, melatonin in children with obesity appears to be involved in the global metabolic and inflammatory alteration of this condition.

## 1. Introduction

Melatonin (N-acetyl-5-methoxytryptamine), the main hormone that regulates circadian rhythms [1,2,3], is secreted by the pineal gland in the evening when blue light decreases, in response to a circadian pacemaker situated in the suprachiasmatic nuclei of the hypothalamus, “the master clock”. Melatonin reaches the peripheral circadian oscillators, acting as a chemical mediator that synchronizes the cellular oscillators in the brain with peripheral organs, and aligning them with external time [4].

The pleiotropy of melatonin to act as a regulator of cell metabolism is related to the diversity in the distribution of its receptors, mainly Melatonin receptor 1 (MT1) and Melatonin receptor 2 (MT2) [3,5,6]. Melatonin has been shown to regulate carbohydrate metabolism through the expression of the glucose transporter (Glut-4) and to participate in immune system regulation [7,8]. In addition to its central role in circadian rhythm regulation, melatonin is also a powerful antioxidant that limits the damage caused by oxidative stress either directly, by scavenging free radicals, or indirectly, by upregulating the expression of antioxidant enzymes or downregulating the expression of enzymes favoring free radical generation [9]. In this context, melatonin can exert anti-inflammatory effects [3,8].

The action of this chronobiotic and cytoprotective agent is dependent on the dynamics of the circulating hormone. Usually, melatonin level increases reaching a peak at the beginning of the night [10,11,12]. Any alteration of the shape or the timing of this peak is considered the most direct way to measure circadian rhythm impairment, also referred to as “chronodisruption” [13,14]. In this way, changes in melatonin levels have been documented in metabolic dysregulation in different animal models [15]. In adult humans, especially in night-shift workers, numerous studies have shown that chronodisruption may contribute to the development of a wide range of disorders, including obesity, hypertension, type 2 diabetes or even immune-mediated diseases and cancers [16,17,18,19]. However, the importance of life habits such as sleep has been investigated in children on only a few occasions [20].

Melatonin is a biomarker used to study the effects of circadian disruption on neurophysiological, and metabolic processes [21]. A variety of methods for sampling and testing melatonin have been described, but there are no established guidelines on the methodology to use. Recently, the analysis of melatonin in saliva samples has emerged as the most practical method currently used in sleep laboratories, but also in studies performed at home [18,21,22].

Therefore, the objective of the present study was to reinforce the knowledge about circadian rhythms in childhood obesity and its relationship with anthropometric parameters, metabolic and inflammation markers. This study was conducted in children with obesity and with normal weight, measuring salivary melatonin that was collected at home at three time points around sleep onset in an easy and feasible way in the pediatric population.

## 2. Materials and Methods

### 2.1. Subjects

The study design is an unmatched grouped case-control–recruitment without sampling. Participants were children 7–14 years old recruited from the outpatient nutrition office of the Pediatrics Department (Dr. Peset University Hospital of Valencia). They were eligible for the study if they met at least one of the following criteria: (1) they were referred for the evaluation of their nutritional status because of excessive weight gain; (2) they were having a familial study of metabolic conditions (hyperlipidemia) or (3) they were going through screening for celiac disease. Children were excluded if they met any of the following criteria: (1) they had a malabsorptive, genetic or endocrinological disease; (2) they had an infection or acute inflammatory response; (3) they were receiving pharmacological treatment; (4) they were receiving melatonin supplementation and (5) were practicing a sport at a high level.

The sample size was calculated according to the following formula: *n* = (2 (Zα + Zβ)^2^ × S^2^)/d^2^. The association level for Zα and Zβ is 99%. As S, we use 11.4, corresponding to the standard deviation of melatonin levels in children, characterized previously [23]. The desired d is 7, corresponding to the minimum difference of melatonin levels in pg/mL. With these parameters there was a total of 64 patients needed per group. Estimating a loss of 15%, 74 patients per group were needed, thus a minimum of 148 participants in total. The initial sample was composed of 267 children, 37 of whom were excluded because they did not perform the minimum saliva collection required for melatonin assessment (100 µL). The definitive sample was comprised of 230 children divided into two groups: a group of children with obesity (the study group) and a group of children with normal nutrition (control group). The study group included 113 children with a body mass index (BMI) percentile >99 for their age and sex, with a mean age of 11.77 years old (standard deviation of 2.39), of which 57 were girls and 56 were boys. The control group included 117 children who attended the consultation for suspected celiac disease or screening for hyperlipidemia with an average age of 11.14 years old (standard deviation of 2.4), of which 59 were girls and 58 were boys.

The study was carried out after the approval of the Ethical Committee. Informed signed consent forms were provided by parents and children older than twelve years old.

### 2.2. Clinical Data

Measurements of weight and height were performed following standardized protocols of the International Society for the Advancement of Kinanthropometry. BMI was calculated as weight in kilograms divided by height in meters squared. BMI z-score values were determined using the World Health Organization tables as references. Body composition was determined by bioelectrical impedance using a Tanita BC-418 MA with 8 contact electrodes (Tanita Europe BV, Hoofddorp, The Netherlands), fat mass percentage was measured, and fat mass index was calculated as fat mass in kilograms divided by height in meters squared.

Systolic and diastolic blood pressure and heart rate were measured by M3 Omron digital blood pressure monitor HEM-7200-E8/(V) (Omron Healthcare, Kyoto, Japan).

The participants were asked to report the time at which they go to bed and the hour at which they wake up, the time at which they have dinner and the number of technological devices present in their bedroom. In addition, they complete with the support of their parents a 3-day recall of their nutritional intake. Analysis of energy and nutrients intake was performed by DIAL software (Alce Ingeniería SA, Madrid, España, http://www.alceingenieria.net/nutricion.htm, accessed on 16 August 2021).

### 2.3. Biochemical Data

Blood samples were collected after twelve hours of fasting. Determinations were performed under standardized conditions at the hospital laboratory. Glucose, high-density lipoprotein cholesterol, apolipoprotein B, aminotransferases and γ-glutamyl transpeptidase were measured using automated direct methods (Aeroset System^®^ Abbott Chemical Clinic, Wiesbaden, Germany). Insulin was determined using automated electrochemiluminescence immunoassay (c8000^®^ Architect, Abbott Clinical Chemistry, Abbott Park, IL, USA). High-sensitivity C-reactive protein was analyzed by immunonephelometry with a Behring 2 nephelometer (Dade Behring, Marbung, Germany).

The adipokines leptin and ghrelin, hormones that trigger satiety and hunger, respectively, and the immune markers monocyte chemoattractant protein-1, plasminogen activator inhibitor-1 (PAI-1), tumor necrosis factor alpha (TNF-α) and interferon-γ were determined via multiplex immunoassay (Labscan 100 Luminex©, Merck Millipore Merck KGaA, Darmstadt, Germany) with Luminex technology using the specific software 3.1 (Luminex Corporation. Austin, TX, USA).

### 2.4. Melatonin Analysis

Three saliva samples were collected from each patient the evening before morning blood extraction. Patients and parents were instructed about the procedure. Questionnaires of sleep habits were distributed to participants that were returned after they filled them. Children were asked not to use technological devices 15–30 min or to perform intense exercise one hour before saliva collection. Each participant went to sleep and woke up at their habitual hour. The collection of saliva according to the habitual bedtime took place under dim light at three time points: 4 h before bedtime, 2 h before bedtime and after 1 h of sleep. The parents were aware of the moment the child fell asleep by the change in respiratory rhythm.

Saliva (at least 100 µL) was collected by the passive drool method into a single-use collection tube. Once collected, the sample was kept at 4 °C and protected from light to prevent melatonin degradation. The sample was retrieved by the investigator the next morning and transferred to a cryotube for congelation at −80 °C until analysis. Before analysis, saliva was centrifuged at 5000× *g* for 15 min at 4 °C. In the supernatant, the melatonin level was quantified via immunoassay using a salivary kit from Salimetrics (Salimetrics, LLC, Carlsbad, CA, USA), and by following the instructions available online at https://salimetrics.com/wp-content/uploads/2018/03/melatonin-saliva-elisa-kit.pdf, accessed on 15 May 2021. The colorimetric reaction was quantified using a VICTOR TM X3 2030 multilabel plate reader (PerkinElmer, Waltham, MA, USA).

### 2.5. Statistics

The statistical analysis was performed with SPSS software (IBM SPSS Statistics for Windows, version 24 (IBM Corp., Armonk, NY, USA)). A Kolmogorov–Smirnov test with Lilliefors correction was used to assess normality of the variables. Since the collected data followed a non-normal distribution, data are expressed as median (interquartile range). The Wilcoxon test for paired data was used for intra-group comparisons, and Mann–Whitney U tests for intergroup comparisons. The correlation assessments between melatonin at 1 h of sleep and the parameters analyzed in children with obesity were performed with Spearman coefficient (rho). A *p* value of <0.05 was considered significant. For the figures, R (R Core Team 2021, https://www.R-project.org/, accessed on 20 September 2021) was used with the packages ppcor [24], dplyr [25] and ggplot2 [26].

## 3. Results

The first time point of melatonin measurement was late afternoon, four hours before bedtime. At this moment, a similar concentration of melatonin was detected in the children of both groups. Two hours later, the melatonin level started to increase in the study group but not in the control group, although no significant difference was detected between them. After going to bed and sleeping for one hour, the parents woke the participants up and helped them to collect the last saliva sample. At this last time point, the melatonin concentration increased in both groups, but the median melatonin concentration in the study group was more than twice the concentration in the control group, and a significant difference was noted with respect to the first point of saliva collection (Table 1 and Figure 1).

Correlation analysis was performed in the study group between melatonin at the last time point (after 1 h of sleep) and the different parameters were analyzed.

Regarding anthropometry, significant correlation was found between melatonin after 1 h of sleep and the BMI z-score (rho = 0.233, *p* = 0.005), fat mass percentage (rho = 0.294, *p* < 0.001) and fat mass index (rho = 0.333, *p* < 0.001). Melatonin concentration at this time point was also significantly correlated with heart rate (rho = 0.288, *p* < 0.001) (Figure 2).

With respect to the metabolic parameters tested, the melatonin concentration after +1 h of sleep only showed an inverse correlation with high-density lipoprotein cholesterol (rho = −0.276, *p* < 0.001). Other parameters related to metabolic risk, such as apolipoprotein B (rho = 0.202, *p* = 0.015), leptin (rho = 0.181, *p* = 0.030) and ghrelin (rho = 0.209, *p* = 0.012) were positively related (Figure 3). Glucose, insulin, aminotransferases and γ-glutamyl transpeptidase were not found to be significantly associated with the nocturnal melatonin level.

When we focused on the relationship between melatonin and inflammatory markers, significant correlation was shown between melatonin after 1 h of sleep and with C-reactive protein (rho = 0.199, *p* = 0.017). Significant association was also observed with the cytokines PAI-1 (rho = 0.204, *p* = 0.014) and TNF-α (rho = 0.229, *p* = 0.006) (Figure 4), but not with monocyte chemoattractant protein-1 nor interferon-γ.

No difference was found between the control and study groups regarding the estimated time to fall asleep (*p* = 0.056), dinner hour (*p* = 0.529), number of technological device (*p* = 0.305) or the caloric intake (*p* = 0.106) and no significant correlation was found between these parameters and the melatonin level 1 h after sleep.

Furthermore, we asked the participants to report the time at which they went to bed and the time at which they woke up. No difference was observed between the two groups at bedtime, but the children with obesity reported an earlier waking hour than the control group, and consequently, the time spent in bed was also shorter (Table 2).

## 4. Discussion

In the present study, we developed a method to determine melatonin levels in salivary samples in an easy manner which may be feasible in children in their usual environment of sleep. Salivary sample collection was performed at three time points around bedtime: two time points in the evening before going to bed and one time point at night after 1 h of sleep. We found a difference in the salivary melatonin concentration between children with obesity and children with a normal weight in the measurement made after one hour of sleep.

These higher melatonin levels may correspond to modifications within the shape, amplitude, or timing of the melatonin peak in children with obesity. The time at which the children were going to sleep and the time to fall asleep was homogeneous between the two groups, whereas the waking up time was earlier in the children with obesity. Thus, the early increase in melatonin level at night could be related to shorter sleep duration. In this sense, it has been reported that insufficient sleep may trigger a metabolic stress response [27].

Although evidence suggests that a late timing of food intake relative to sleep time could have a negative impact on obesity [28], in the current study no difference in the time of dinner and time of sleep was found between children with obesity and children with normal BMI.

In the context of obesity, sleep deprivation may be an aggravating factor of cardiovascular risk because it can affect cardiac physiology, lipid metabolism and inflammation. High heart rate is known to be correlated to fat mass and BMI [29]. Studies in adolescents and young males suggests that an increased percentage of body fat is associated with reduced cardiac parasympathetic and increased sympathetic activity [30,31]. We have observed that melatonin level at 1 h after sleep was related to the heart rate in children with obesity. This is in line with animal studies that found that increased sympathetic tone might underly a potentially compensatory increase in melatonin concentration [32].

Higher nocturnal melatonin levels were also found in children with lower values of high-density lipoprotein cholesterol, a known protective factor for metabolic diseases. Other parameters also correlated with melatonin levels were markers of subtle metabolic disturbance, as apolipoprotein B, a classical marker of cardiovascular risk and considered a proatherogenic factor [33]. The associations found between the night melatonin and these different factors could support the hypothesis of a potential atheroprotective role of melatonin which is increased to counteract the atherogenic effects of obesity. Whereas animal models provide evidence of the beneficial regulation of cholesterol by exogenous melatonin [34], clinical observations mainly failed to find a protective effect [35,36,37,38]. Although this difference may be explained by the high interindividual variability in the bioavailability of melatonin in humans [39], such inconsistencies make it difficult to give a reliable opinion on the action of melatonin on the metabolism of lipoproteins.

Likewise, the orexigenic and anorexigenic hormones ghrelin and leptin, characteristically affected by obesity, were also found to be correlated with melatonin at night. Melatonin levels may affect the secretion and activity of leptin and ghrelin [40]. Indeed, a shorter sleep duration implicates a greater period of wakefulness, which results in increase in energy consumption. To compensate for this effect and restore the energetic balance, melatonin may modulate the nutritional intake through regulation of the hormones of hunger and satiety.

Finally, melatonin levels after 1 h of sleep were found to be positively associated with inflammatory markers, high-sensitivity C-reactive protein, PAI-1 and TNF-α. Obesity is considered a chronic inflammatory state of moderate intensity, frequently referred to as “low grade and chronic inflammation” [41,42,43], of which the mentioned markers are characteristic [44,45,46,47].

According to the results presented, we suggest that melatonin increase may represent a counterregulatory mechanism against the consequences of obesity and the stimulation of its endogenous production might attenuate the associated alterations.

The results of the present work may seem paradoxical because melatonin is habitually presented as a protecting agent against metabolic disorders and obesity in intervention studies. Indeed, animal studies performing pinealectomy and melatonin supplementation showed a BMI reduction. Similarly, melatonin supplementation in human studies showed beneficial effects on blood pressure and inflammation. Additionally, several studies suggest that low melatonin production is associated with a higher risk of cardiac illnesses such as left ventricular hypertrophy, coronary heart disease and congestive heart failure [13,48].

However, some studies also report, similar to the current study, an increase in melatonin levels in patients with obesity [49]. In a study carried out in patients with cranial tumors, melatonin levels at night were found to be proportional to BMI [50]. It is possible that evolution of obesity over time may reveal the true meaning of the increased melatonin in situations of cardiovascular risk and inflammation.

Limitations of the present study should be mentioned. No sleep monitoring device was used. Therefore, no objective measurement proves the timing of collection respective to sleep times. The parents were rigorously committed to determine when the child fell asleep. This aspect could likely to limit the bias in a certain proportion. It also should be noted the main strength of this study that lies in the melatonin measurement in the habitual environment of the children facilitating the inclusion of a relatively high number of participants.

## 5. Conclusions

We have found an increase in melatonin levels at 1 h of sleep in children with obesity, related to fat mass and cardiometabolic risk factors. Melatonin, involved in antioxidant and anti-inflammatory defense, might be upregulated in the context of obesity as a compensatory mechanism. The organism would trigger its production to increase sleepiness and favor behaviors towards gaining sleep time or counteract the pro-inflammatory and oxidative stress effects of obesity and sleep deficiency.

## Figures and Tables

**Figure 1 nutrients-13-03629-f001:**
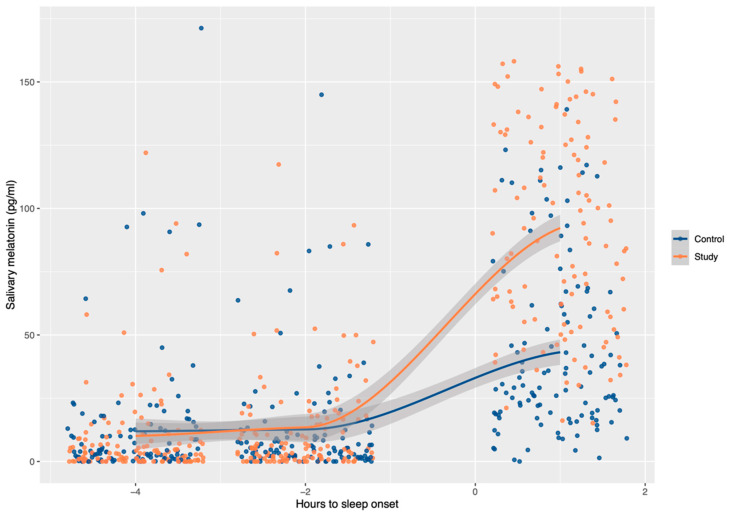
Evolution of salivary melatonin during the evening and night in children from the study and control groups. At the two time points before sleep onset, melatonin levels were equivalent in children from both groups. After one hour of sleep, the children with obesity presented a significantly higher melatonin level than the controls (*p* < 0.001). The curves bind the medians of the three measurements using a locally weighted polynomial (LOESS).

**Figure 2 nutrients-13-03629-f002:**
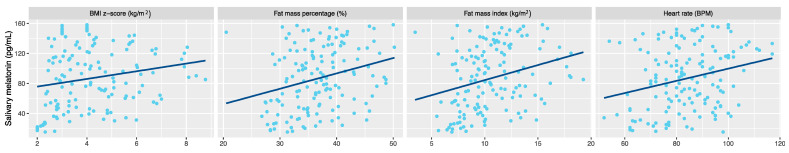
Correlation between night melatonin level (after 1 h of sleep) and anthropometric and clinical parameters in children with obesity. Body mass index-z-score (BMI z-score). Beats per minute (BPM).

**Figure 3 nutrients-13-03629-f003:**
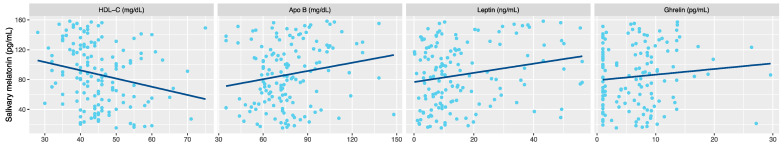
Correlation between night melatonin level (after 1 h of sleep) and biochemical parameters in children with obesity. High-density lipoprotein-cholesterol (HDL-C). Apolipoprotein B (Apo B).

**Figure 4 nutrients-13-03629-f004:**
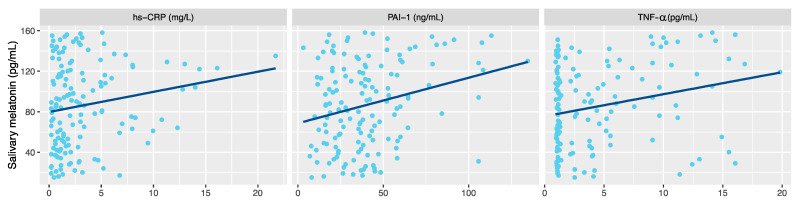
Correlation between night melatonin level (after 1 h of sleep) and inflammatory parameters in children with obesity. High-sensitive c-reactive protein (hs-CRP). Plasminogen activator inhibitor-1 (PAI-1). Tumor necrosis factor-α (TNF-α.).

**Table 1 nutrients-13-03629-t001:** Salivary melatonin levels (pg/mL) in children with obesity and control children during the period of study.

Time Relative to Sleep Onset	Study Group	Control Group	*p*-Value
−4 h	3.55 (0.65–9.23)	3.72 (0.97–12.60)	0.659
−2 h	5.13 (1.22–17.75)	5.35 (1.84–12.23)	0.930
+1 h	90.16 (57.16–129.16) *	29.82 (19.05–61.54) *****	<0.001

Data are presented in median (interquartile range). Comparisons intergroup were made by Mann–Whitney U tests. Comparison intragroup was made by paired Wilcoxon test for paired data. * *p* < 0.001 vs. −2 h and −4 h.

**Table 2 nutrients-13-03629-t002:** Self-reported timing of sleep, in children with obesity and control children.

	Study Group	Control Group	*p*-Value
Bedtime (h)	22:30 (22:00–23:00)	22:30 (22:00–23:00)	0.166
Wake time (h)	7:15 (7:00–7:57)	7:30 (7:00–8:00)	<0.001
Nighttime sleep (h)	8.45 (8.00–9.30)	9.00 (8.25–10.00)	0.002

Data are presented in median (interquartile range). Comparisons intergroup were made by Mann–Whitney U tests.

## Data Availability

The data presented in this article are available on reasonable request from the corresponding author.

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
