# Peer review of "Melatonin Levels in Children with Obesity Are Associated with Metabolic Risk and Inflammatory Parameters"

_nutrients, 2021, doi:10.3390/nu13103629_

Round 1

Reviewer 1 Report

The paper is interesting, however too long. 

To short it about 1/4.

To add the important numbers describing results,  in the abstract.

To indicate the limitation of the study. 

To modify the conclusions. I propose to conclude as the wide, short summary of the paper instead repeating the results.

Author Response

We thank the reviewer for the valuable comments.

  1. The paper is interesting, however too long. To short it about 1/4.

Response: We have shortened the article by approximately 500 words. We believe that the remaining text is necessary to understand the article.

.To add the important numbers describing results, in the abstract.

Response: Specific results are added in the abstract on lines 19-21

  1. To indicate the limitation of the study.

Response: Limitation is described (page 7, lines 263-265)

  1. To modify the conclusions. I propose to conclude as the wide, short summary of the paper instead repeating the results.

Response: The conclusions have been modified according to your suggestion (page 7, line 272-276)

 We thank you again for your insight and hope that the changes we did satisfy your requests

Reviewer 2 Report

This is a very strong manuscript however significant flaws in the study design undervalue its merit for publication. If authors responce adequately to the comments. Otherwise the flwas in their study eliminate the robusteness of this research. 

Author Response

The authors appreciate the comments of the reviewer and have modified the article according them. The flow of the article has been improved. The authors hope that the questions proposed by the reviewer have been addressed and adequately responded.

Response: We have modified the introduction, and selected references were included. Study design has been explained. Methods and results have been clarified. Conclusion has been modified

Reviewer 3 Report

The study measured melatonin levels in saliva of children 7–14 years old with obesity >99 percentile, and non-obesity. Although the melatonin levels were not different before sleep between obese, and control subjects, they were different one hour of sleep. High-density lipoprotein cholesterol, alanine transaminase, γ-glutamyl transpeptidase, leptin, ghrelin, MCP-1, PA-1, TNF-α, INF-γ in morning blood specimens were positively or negatively associated with melatonin levels one hour of sleep in obese subjects. From the cross-sectional observational study, the authors concluded that “children with obesity also present higher levels of melatonin at night, suggesting a compensatory mechanism,” and “the increase in melatonin at night in children with obesity appears to be involved in the global metabolic and inflammatory alteration of this condition.” It is very disappointing as one of the readers that the study did not measure or mention nutrients.

The authors described that they developed a methodology to assess melatonin levels in saliva at three time points around sleep onset: 4, and 2 hour before sleep, and 1 hour after sleep. This method must be a user-friendly manner and feasible for children (Lines 225, 226). Other studies collect saliva over a 24 h time period with every 2–3 h to determine DLMO (Line 228). But the authors did not validate whether the method with three time points corresponds to that with 2-3 h intervals over a 24 h time period. The authors did not record the time to fall in sleep and did not measure the quality of sleep. Melatonin levels at the third point cannot represent that 1 h of sleep because they did not use a sleep monitor.

There is no rationale to measure hormones such as leptin, and ghrelin, and immune markers such as MCP-1, PAI-1, TNF-α, INF-γ in the Introduction and the Methods. These parts are not needed.

Although the authors suggested global metabolic and inflammatory alteration due to melatonin with antioxidant or other roles, they did not directly measure or analyze these effects of melatonin in obese subjects.

Partial correlation coefficients are too small, and the parameters have not normal distributions.

References are not enough.

Author Response

The authors are grateful for the constructive comments of the reviewer about the article. They have been carefully recorded and incorporated to the text. We thank you again for your insight and hope that the changes we did satisfy your requests

  1. The study measured melatonin levels in saliva of children 7–14 years old with obesity >99 percentile, and non-obesity. Although the melatonin levels were not different before sleep between obese, and control subjects, they were different one hour of sleep. High-density lipoprotein cholesterol, alanine transaminase, γ-glutamyl transpeptidase, leptin, ghrelin, MCP-1, PA-1, TNF-α, INF-γ in morning blood specimens were positively or negatively associated with melatonin levels one hour of sleep in obese subjects. From the cross-sectional observational study, the authors concluded that “children with obesity also present higher levels of melatonin at night, suggesting a compensatory mechanism,” and “the increase in melatonin at night in children with obesity appears to be involved in the global metabolic and inflammatory alteration of this condition.” It is very disappointing as one of the readers that the study did not measure or mention nutrients.

Response: Our results seem contrary to other related in the literature. However , it can be correspond to different moments of the evolution of obesity similarly to the process of insulin resistance. In a first time (in childhood) there is a compensatory increase that evolves to a diminution at a long-time. This is an aspect to study in future works.

Respecting to dietetic assessment, we have made the statistical analysis of the children diet that was systematically recorded and no relationship between melatonin and energy or nutrients was found.

  1. The authors described that they developed a methodology to assess melatonin levels in saliva at three time points around sleep onset: 4, and 2 hour before sleep, and 1 hour after sleep. This method must be a user-friendly manner and feasible for children (Lines 225, 226). Other studies collect saliva over a 24 h time period with every 2–3 h to determine DLMO (Line 228). But the authors did not validate whether the method with three time points corresponds to that with 2-3 h intervals over a 24 h time period. The authors did not record the time to fall in sleep and did not measure the quality of sleep. Melatonin levels at the third point cannot represent that 1 h of sleep because they did not use a sleep monitor.

Response: We agree to the reviewer. That is the main limitation of the study. But it was rigorously carried out to determine the time of the child fell asleep. Parents were instructed carefully about the signs that they should observe.

  1. There is no rationale to measure hormones such as leptin, and ghrelin, and immune markers such as MCP-1, PAI-1, TNF-α, INF-γ in the Introduction and the Methods. These parts are not needed.

Response: The authors believe that these biomarkers are part of the inflammatory process that accompanies obesity and they consider important to include them

  1. Although the authors suggested global metabolic and inflammatory alteration due to melatonin with antioxidant or other roles, they did not directly measure or analyze these effects of melatonin in obese subjects.

Response: You are right. We have a study in progress about antioxidant role of melatonin. It is a hypothesis in the present article that the role of melatonin could be mediated by its antioxidant action.

  1. Partial correlation coefficients are too small, and the parameters have not normal distributions.

Response: We agree. We have modified the statistical analysis. Because the data have a non-normal distribution, non-parametric tests were used.

  1. References are not enough.

Response: We have increased the number of references according to your suggestion

Round 2

Reviewer 2 Report

Dear Authors, 

due to my mistake in the copy-paste procedure, you did not received my comments to you but the coments to the Editors thus no specific change was done in the manuscript. My comments for your manuscript are the following, in order to address them and afterward I will proceed with the proper evaluation of your manuscirpt. 

This is a very interesting manuscript, assessing the relation of an important and really under- researched hormone such as melatonin in children. Since there are evidence about the relation between sleep duration and sleep quality with childhood obesity, the exploration of possible pathophysiological mechanisms about this relation is mandatory. Thus, this manuscript is of great importance. However, there are some flaws in the study design that decrease the robustness of the finding of the authors:  

Major comments:

Material and methods, line 78-79: The study design is not described well. Authors say that this is a cross-sectional design but it seems that the design is unmatched grouped case-control – recruitment without sampling and division of the original sample into cases (obese children and non-obese controls). Authors should make the appropriate amendments thought the relevant sections. Moreover, no sample size calculations are provided. Authors should document about the power of their sample. Finally, if one of the entry criteria for the study is for a child to have been referred for excessive weight gain, how it is possible to gather a normal weight group of subjects? Authors should provide more details in order to explain this discrepancy.

Results: Authors should not use p=0.000. The correct expression is p<0.001 Moreover, since the regression lines are presented, it would be more informative to present linear regression coefficients also in their results or should document why they used only partial correlation coefficient analyses.

Furthermore, important confounders are lacking from their analysis, such as socioeconomics, dietary and physical activity variables. Authors should document more about the validity of their findings in terms of dealing with confounding.

Minor comments: Abstract: Some estimators would be useful to be presented in the abstract.

Author Response

Reviewer #2

Dear Authors, 

Due to my mistake in the copy-paste procedure, you did not received my comments to you but the coments to the Editors thus no specific change was done in the manuscript. My comments for your manuscript are the following, in order to address them and afterward I will proceed with the proper evaluation of your manuscript. 

This is a very interesting manuscript, assessing the relation of an important and really under- researched hormone such as melatonin in children. Since there are evidence about the relation between sleep duration and sleep quality with childhood obesity, the exploration of possible pathophysiological mechanisms about this relation is mandatory. Thus, this manuscript is of great importance. However, there are some flaws in the study design that decrease the robustness of the finding of the authors:  

Response: Thank you very much for your kind comments. We have appreciated them very much.

Major comments

  1. Material and methods, line 78-79: The study design is not described well. Authors say that this is a cross-sectional design but it seems that the design is unmatched grouped case-control – recruitment without sampling and division of the original sample into cases (obese children and non-obese controls).

Response: We made the modification according to your suggestion, as you will find in line 69.

  1. Authors should make the appropriate amendments thought the relevant sections. Moreover, no sample size calculations are provided. Authors should document about the power of their sample.

Response: Thank you for this valid point, we included the calculations on the lines 77-84.

  1. Finally, if one of the entry criteria for the study is for a child to have been referred for excessive weight gain, how it is possible to gather a normal weight group of subjects? Authors should provide more details in order to explain this discrepancy.

Response: We believe that there is a misunderstanding and hope we made it clearer in the text, line 70 to 74. Indeed, the children eligible for the study were those presenting at least one of the criteria and finally, the participants included in study group were the children with obesity (>99% BMI percentile) and the control group were children who attended the consultation for suspected celiac disease or screening for hyperlipidemia.

  1. Results: Authors should not use p=0.000. The correct expression is p<0.001.

Response: The changes have been done accordingly, thank you for your comment.

  1. Moreover, since the regression lines are presented, it would be more informative to present linear regression coefficients also in their results or should document why they used only partial correlation coefficient analyses.

Response: During the first round of the reviewing process, we have made several changes and selected the most adapted statistical tests.

  1. Furthermore, important confounders are lacking from their analysis, such as socioeconomics, dietary and physical activity variables. Authors should document more about the validity of their findings in terms of dealing with confounding.

Response: It is correct. Neither the caloric content of the diet, time of meals, exposure to technological devices, and estimated time to fall asleep results were found different between the groups, therefore we did not include them in the first version of the article. Nevertheless, they have been included some of these in a second time, in the results lines 205-208. Due to the short time provided  by the Editor (4 days) for this second revision we have not been able to analyze all remaining variables that have been collected.

Minor comments

  1. Abstract: Some estimators would be useful to be presented in the abstract.

Response: It has been done as well.

Reviewer 3 Report

The authors did not address the comments.  Melatonin was not measured based on the time children fell in sleep. Metabolic parameters are not related with melatonin levels, and obesity is the main cause, so the causality relationship is broken. 

Author Response

Reviewer #3  

  1. The authors did not address the comments. Melatonin was not measured based on the time children fell in sleep.

Response: Dear colleague, we apologized if we made you feel like we did not take your comment in consideration. In fact, we have consulted the sleep unit of our university hospital, who assured us that the only incontestably reliable way to assess is an individual is asleep is the use of electroencephalography. To date, this can be performed exclusively in specialized laboratories, which is incompatible with the present study’s objective that is to assess melatonin in children in their natural sleep environment. We understand that the measurement could be not performed at an accurate timing regarding to sleep onset, and you were right that it should be written clearly as a limitation, as it is done now. If we focus on the bias this may cause that the saliva collection for melatonin assessment may happen either too early, or too late, reflecting a different point of the peak of melatonin either too low or too high. The melatonin is of endocrine nature with a relatively slow evolution, with a peak often reported around 60 pg/mL and lasting approximately 4 hours to reach the maximum. Indeed, if an error of 15 minutes was done in the collection time, the difference would be of 3,75 pg/mL. We believe that due the range of values, the median of the results and the relatively high number of participants, the mesures are likely to be representatives enough of the groups studied.

  1. Metabolic parameters are not related with melatonin levels, and obesity is the main cause, so the causality relationship is broken.

Response: You are absolutely right, the present study design does not permit to measure causality, only associations. This is why we are particularly cautious in mentioning exclusively “relationships” and “associations”, and never in a manner which may imply causality. The conclusion proposed is that melatonin may be one of de various factors which levels are modified in the context of childhood obesity. Moreover, as melatonin is usually presented as a molecule with virtuous effects on metabolic health, our hypothesis is that this change in physiology could eventually be a strategy of the organism to counteract the deleterious effects of inflammation and metabolic unbalance in obesity.